# Investigating Polymorphisms and Expression Profile of Immune, Antioxidant, and Erythritol-Related Genes for Limiting Postparturient Endometritis in Holstein Cattle

**DOI:** 10.3390/vetsci10060370

**Published:** 2023-05-23

**Authors:** Mona Al-Sharif, Mohamed Abdo, Omnia El Shabrawy, Eman M. Abu El-Naga, Liana Fericean, Ioan Banatean-Dunea, Ahmed Ateya

**Affiliations:** 1Department of Biology, College of Science, University of Jeddah, Jeddah 21589, Saudi Arabia; mmalshreef@uj.edu.sa; 2Department of Animal Histology and Anatomy, School of Veterinary Medicine, Badr University in Cairo (BUC), Cairo 11829, Egypt; mohamed.abdo@vet.usc.edu.eg; 3Department of Anatomy and Embryology, Faculty of Veterinary Medicine, University of Sadat City, Sadat City 32897, Egypt; 4Department of Biochemistry and Chemistry of Nutrition, Faculty of Veterinary Medicine, Menofia University, Menofia 32951, Egypt; omnia.ibrahim@vet.menofia.edu.eg; 5Department of Theriogenology, Faculty of Veterinary Medicine, Aswan University, Aswan 81528, Egypt; e.abuelnaga@vet.aswu.edu.eg; 6Department of Biology and Plant Protection, Faculty of Agricultural Sciences, University of Life Sciences King Michael I, 300645 Timisoara, Romania; ioan_banatean@usab-tm.ro; 7Department of Development of Animal Wealth, Faculty of Veterinary Medicine, Mansoura University, Mansoura 35516, Egypt

**Keywords:** Holstein cattle, genetic polymorphisms, transcript level, postparturient endometritis

## Abstract

**Simple Summary:**

Different genetic loci have a significant impact on individual susceptibility to various bacterial infections, which may help to explain the unique phenotypic presentation of postpartum endometritis. Finding the genes and mutations that cause the variation in disease resistance could greatly improve the efficacy of breeding animals with innate disease resistance. Molecular genetic analyses of the immunological (*TLR4*, *TLR7*, *TNF-α*, *IL10*, *NCF4*, and *LITAF*), antioxidant (*ATOX1*, *GST*, and *OXSR1*), and erythritol-related (*TKT*, *RPIA*, and *AMPD1*) genes comparing healthy and endometritis cows found differences in nucleotide sequence and transcript levels. This remark might indicate that healthy animals have their immune systems well controlled. These genes’ abundance in transcripts offers a possible source of postpartum uterine health markers.

**Abstract:**

This study looked at genetic polymorphisms and transcript levels of immune, antioxidant, and erythritol-related markers for postparturient endometritis prediction and tracking in Holstein dairy cows. One hundred and thirty female dairy cows (65 endometritis affected and 65 apparently healthy) were used. Nucleotide sequence variations between healthy and endometritis-affected cows were revealed using PCR-DNA sequencing for immune (*TLR4*, *TLR7*, *TNF-α*, *IL10*, *NCF4*, and *LITAF*), antioxidant (*ATOX1*, *GST*, and *OXSR1*), and erythritol-related (*TKT*, *RPIA*, and *AMPD1*) genes. Chi-square investigation exposed a noteworthy variance amongst cow groups with and without endometritis in likelihood of dispersal of all distinguished nucleotide variants (*p* < 0.05). The *IL10*, *ATOX1*, and *GST* genes were expressed at substantially lower levels in endometritis-affected cows. Gene expression levels were considerably higher in endometritis-affected cows than in resistant ones for the genes *TLR4*, *TLR7*, *TNF-α*, *NCF4*, *LITAF*, *OXSR1*, *TKT*, *RPIA*, and *AMPD1*. The sort of marker and vulnerability or resistance to endometritis had a significant impact on the transcript levels of the studied indicators. The outcomes might confirm the importance of nucleotide variants along with gene expression patterns as markers of postparturient endometritis susceptibility/resistance and provide a workable control plan for Holstein dairy cows.

## 1. Introduction

The three weeks before and after parturition, or the periparturient phase, are when an animal experiences the largest alterations to its endocrine system [1]. These changes are significantly more pronounced than they are at any other time during the lactation–gestation cycle [2]. Because of the decreased feed intake, endocrine, and metabolic fluctuations after parturition, the postpartum period is the most worrying time [3]. It is also frequently accompanied by a number of physiological stresses that have an immunological source. Given that nearly 75% of diseases in adult dairy cows normally manifest within the first month following delivery [4], the first 10 days following calving are when the overall number of diseases—including mastitis, ketosis, endometritis, digestive problems, and lameness—is most likely to develop [5]. According to some theories, periparturient diseases in dairy cows are brought on by an imbalance between the body’s insufficient antioxidant protection and amplified synthesis of lipid peroxides and reactive oxygen species (ROS) [6].

*Bos taurus* and chiefly dairy cattle are disposed to uterine infection and illness post-delivery [7]. Uterine involution, which is marked by the exclusion of bacterial infection and rejuvenation of the endometrial tissue, is crucial for recurring uterine receptiveness and founding conception in postpartum dairy cows [8]. Cows can battle to quickly eliminate uterine infections due to unfortunate immune response [9]. This is connected to an ongoing inflammatory response in endometrial tissue, which is where endometritis originates [10]. In the period after parturition, dairy cows are liable to bacterial uterus infections [11]. Since a variety of bacteria may be easily isolated from the uterine lumen, it is believed that bacterial infections are the chief origin of the majority of uterine diseases [7,12]. Due to poor inflammation activation and bacterial clearance, endometritis has also been linked to augmented levels of proinflammatory cytokine transcripts [13]. In dairy cattle, uterine disorders (UD) are very common and cause noteworthy economic losses [14]. Metritis (MET), an acute inflammatory disorder distressing all layers of the uterine wall within 21 to 50 days following delivery, is estimated to cost USD 350 per case by Overton and Fetrow [15].

Marker-aided selection (MAS) is a method for identifying genetic variables that influence susceptibility to common diseases [16,17]. In the era of genomic selection, the implementation of selection approaches for novel functional qualities, such as disease resistance, is made possible by large cow training sets merging phenotypes with high-throughput genomic SNP indicator information [18]. The term “transcriptome” denotes is the full range of messenger RNA, or mRNA, molecules expressed by an organism [19]. It is frequently used to immunological monitoring in inflammatory disorders to locate pathogenic, diagnostic, and prognostic indications. It is essential for discovering new therapeutic or diagnostic targets [20,21]. Antibiotic residues and preventive medications are no longer desirable due to rising disease resistance [22]. The answer to low-cost and effective control of these diseases is to take advantage of host genetic resistance because it is not constrained by these drawbacks in the broadest sense [23]. Unfortunately, numerous gene loci need to be found and specified in order to effectively manage these illnesses [24].

Several regulatory enzymes of the intermediary metabolism have variable gene expression, which can offer helpful methods for enhancing genetic selection for cattle adaptation to adverse environments [25]. Because there is an excess of erythritol in the tissues of bovine fetuses, most bacteria have the unusual facility to catabolize erythritol, which has been linked to their pathogenicity [26]. Since erythritol was identified as the cause of B. abortus localization in the placenta of pregnant cows, it has been assumed that erythritol contributes to Brucella pathogenicity [26]. Ribose 5-phosphate isomerase A (RPIA) and transketolase (TKT), which are involved in glycerol metabolism and boosting erythritol manufacture, were co-overexpressed and significantly augmented erythritol development [27]. The AMP deaminase-encoding gene (*AMPD*) is advantageous for erythritol production and governs carbon fluidities in glycolysis and the TCA cycle [28]. The relationship between erythritol-related genes and endometritis susceptibility in Holstein dairy cows has not been extensively studied previously. Diagnoses and management of complex disorders under a multigenic regulator are more challenging than those under a unigenic one [29]. Numerous candidate gene variations and anonymous markers are searched for connections with disease resistance in order to identify disease-resistance indicators. Quantitative trait loci (QTLs) are then mapped [30]. Only modest progress has been made in determining the molecular genetic roots of endometritis in animals [31,32]. This study used PCR-DNA sequencing and real-time PCR to examine potential immune (*TLR4*, *TLR7*, *TNF-α*, *IL10*, *NCF4*, and *LITAF*), antioxidant (*ATOX1*, *GST*, and *OXSR1*), and erythritol-related (*TKT*, *RPIA*, and *AMPD1*) gene efficacy as candidates for prediction and tracking endometritis resistance/susceptibility in postparturient Holstein dairy cows.

## 2. Material and Methods

### 2.1. Dairy Cows and Research Samples

This research used Holstein dairy cows (*n* = 130) reared on a private farm in the area of Ismailia, Egypt, of which 65 were endometritis-affected and 65 appeared to be in good health. All cows were inspected by the same veterinarian each day. The endometritis cows were selected based on body temperature and physical examination findings during postparturient period (40 to 60 days postpartum), with close consideration to body temperature. The animals were examined and findings (body temperature, pulse, respiration rate, mucous membranes, and vaginal discharge) recorded [33]. The first group involved clinically well-fitted cows that had had a usual calving and standard postpartum stage (i.e., customary feed consumption, body temperature, no uterine discharge). The second group included cows indicating endometritis (pyrexia, tenacious-colored uterine discharge with offensive odor, anorexia, depression).

The cows, in their third lactation season, were raised in a commercial dairy herd of roughly 500 animals. Cows normally weighed 470 kg and were 5 years old. The livestock were kept in a cubicle (free-stall/feedlot) barn that featured straw-lined stalls, a slatted floor that was habitually scraped, a total mixed ration (TMR), twice-daily milking, and artificial insemination. The jugular vein of every cow was punctured to attain five milliliters of blood. In order to recover DNA and RNA, the samples were positioned into tubes filled with anticoagulants in a vacuum to acquire whole blood (EDTA or sodium fluoride). All animal management procedures, tested trial gathering, and sample discarding were carried out underneath the supervision of the University of Sadat City’s Veterinary Medical School in accordance with IACUC guidelines (code VUSC-015-1-23).

### 2.2. Isolation and Amplifying DNA

By means of the genetic material JET entire blood genomic DNA isolation kit and the manufacturer’s guidelines, total blood was used to retrieve the genome’s DNA (Thermo scientific, Vilnius, Lithuania). DNA with a high degree of purity and concentration was examined using Nanodrop. Immune (*TLR4*, *TLR7*, *TNF-α*, *IL10*, *NCF4*, and *LITAF*), antioxidant (*ATOX1*, *GST*, and *OXSR1*), and erythritol-related (*TKT*, *RPIA*, and *AMPD1*) genes were amplified. The *Bos taurus* genome accessible in PubMed was employed to create the oligonucleotide sequences for amplification. Table 1 contains a list of the primers used during the PCR.

A heat cycler with a 150 mL final bulk was used to process the polymerase chain amplification mixture. Each reaction container contained the following components: 66 μL d.d. water, genetic material with 6 microliters, each matching primer with 1.5 microliters, and of master mixture with 75 microliters (Jena Bioscience, Jena, Germany). At a beginning 95 °C for unwinding temperature, the PCR combinations stayed spent four minutes. The 35-cycles included 95 °C denaturation cycles lasting one minute each, annealing cycles lasting one minute at the temps listed in Table 1, 30 s rounds for elongation at 72 °C; ten additional minutes of extending occurred at 72 °C. The materials were saved at 4 °C. A gel certification system was employed to find demonstrative PCR findings using agarose gel electrophoresis and to view PCR segment patterns under UV light.

### 2.3. Finding Polymorphism

Prior to DNA sequencing, Jena Bioscience # pp-201s/Munich, Hamburg, Germany, offered tools for purifying PCR and eliminating primer dimmers, nonspecific bands, and other contaminants, producing the intended amplified product of the predicted scope [34]. To measure PCR output, satisfactory quality and good concentrations were achieved by employing a Nanodrop (Waltham, MA, USA, UV-Vis spectrometer Q5000) [35]. Healthy alongside endometritis-affected cows were used to search for SNPs using sequencing of the amplified products containing the actual PCR result. The PCR yields were sequenced on an ABI 3730XL DNA sequencer (United States: Applied Biosystems, Waltham, MA, USA) operating the Sanger et al. [36]-described enzyme chain terminator method.

The software programs Chromas 1.45 and BLAST 2.0 were used for examining the DNA analysis outcomes [37]. Polymorphisms were identified when comparing the immune, antioxidant, and erythritol-related gene products produced using PCR to the reference gene sequences obtained from GenBank. Relying on the sequence matching amongst the dairy cows, the MEGA4 tool has the ability to recognize dissimilarities in the examined genes’ amino acid sequences [38].

### 2.4. Transcript Levels of Immune, Antioxidant and Erythritol Related Genes 

Following the manufacturer’s guidelines, the whole RNA was extracted from the blood samples taken from the investigated dairy cows using the Trizol solution (RNeasy Mini Ki, 74104, Qiagen, Venlo, The Netherlands, 74004). Using a NanoDrop^®^ ND-1000 spectrophotometer, we quantified and confirmed the amount of the extracted RNA. The producer’s technique was utilized for producing the complementary nucleic acid for every sample (Waltham, MA, USA: Thermo Fisher, product no. EP0441). SYBR Green PCR Master Mix and quantifiable RT-PCR were employed to evaluate the expression profiles of immune (*TLR4*, *TLR7*, *TNF-α*, *IL10*, *NCF4*, and *LITAF*), antioxidant (*ATOX1*, *GST*, and *OXSR1*), and erythritol-related (*TKT*, *RPIA*, and *AMPD1*) genes (2x SensiFastTM SYBR, Bio-line, CAT No: Bio-98002). The SYBR Green PCR Master Mix was exploited for calculating comparative amount possessed by the mRNA (Toronto, ON, Canada: Quantitect SYBR green PCR reagent, catalog no. 204141).

The sequences for sense as well as antisense primers were created using the *Bos taurus* genome found in PubMed (Table 2). The *ß. actin* gene served as the constitutive normalization reference. Overall RNA with 25 microliters, 1 microliter of every matching primer, 8 microliters of water without nuclease, 0.5 microliters of reverse transcriptase, 12.5 microliters of Quantitect SYBR green reaction master solution, and 3 microliters of Trans Amp buffer made up the PCR combination. The finished reaction mixture then underwent the following steps inside a heater cycler: inverse transcription for 30 min at 55 °C; preliminary denaturation aimed at 8 min at 95 °C; 40 cycles at 95 °C aimed at 15 s and the primer binding temperatures specified throughout Table 2; and extending aimed at 1 min at 72 °C. A melting curve investigation was employed subsequent to the amplifying step for proving specificity of the amplified product. By comparing each gene’s expression in the analyzed sample to that of the *ß. Actin* gene, the 2^−ΔΔCt^ scheme was exploited for considering the differences in the expression of each gene [39,40].

### 2.5. Statistical Analysis

**H_o_:** 
*Polymorphisms and expression profile of immune, antioxidant, and erythritol-related genes could not limit postparturient endometritis in Holstein cattle.*


**H_A_:** 
*Polymorphisms and expression profile of immune, antioxidant, and erythritol-related genes limited postparturient endometritis in Holstein cattle.*


The substantial differences in the discovered genes’ SNPs between the examined cows were found using a chi-square analysis. A statistical investigation was exploited for this reason using the GraphPad statistical program (*p* < 0.05). The *t*-test and form 17 of the statistical software set Statistical Program for Social Science (SPSS) was exploited for judging if it was present a statistically noteworthy variance between healthy and endometritis-affected cows. Mean and standard error (mean ± SE) were used to present the findings. The significance of the variations was assessed using *p* < 0.05. The investigated immune, antioxidant, and erythritol-related genes’ transcript levels served as an unchanging factor for designating healthy and endometritis-affected cows as the reliant factor, and significance of the numerous influences was judged using a distinguishable investigation model. To differentiate between endometritis-affected and healthy cows, the transcript quantities of indicators undergoing examination were utilized. A two-way ANOVA and a univariate general linear model (GLM) were used for investigating the relationship between two variables (indicator kind alongside endometritis incidence) and how it impacts transcript levels. 

## 3. Results

### 3.1. Genetic Polymorphisms of Immune, Antioxidant, and Erythritol-Related Genes 

The PCR-DNA sequence verdicts of healthy and affected dairy cows revealed differences in the SNPs in the amplified DNA bases related to endometritis for the *TLR4* (528-bp), *TLR7* (420-bp), *TNF-α* (551-bp), *IL10* (571-bp), *NCF4* (865-bp), *LITAF* (644-bp), *ATOX1* (450-bp), *GST* (480-bp), *OXSR1* (525-bp), *TKT* (456-bp), *RPIA* (390-bp), and *AMPD1* (502-bp) genes. All the discovered SNPs were approved using the DNA sequence differences between immune, antioxidant, and erythritol-related markers investigated in the researched cows and the reference gene sequences obtained from GenBank (Appendix A). The healthy and endometritis-affected cows exhibited noticeably altered incidences of the studied markers, as determined using the SNPs’ chi-square analysis (*p* < 0.05) (Table 3). The exonic region changes shown in Table 1 were present in all of the immune, antioxidant, and erythritol-related markers under investigation, causing coding DNA sequence alterations in the affected cows compared to healthy ones.

### 3.2. Patterns for Transcript Levels of Immune, Antioxidant, and Erythritol-Related Indicators

In Figure 1, the transcript profiles for the assessed immune, antioxidant, and erythritol-related indicators are displayed. The *IL10*, *ATOX1*, and *GST* genes were expressed at substantially lower levels in endometritis-affected cows. Gene expression levels were considerably higher in endometritis-affected cows than in resistant ones for the genes *TLR4*, *TLR7*, *TNF-α*, *NCF4*, *LITAF*, *OXSR1*, *TKT*, *RPIA*, and *AMPD1*. 

The sort of indicator and vulnerability or resistance to endometritis had a significant impact on the mRNA concentrations of the indicators being studied. For each gene examined in the endometritis-affected cows, *TLR4* had the highest potential mRNA level (2.51 ± 0.11); *IL10* had the lowest potential level (0.31 ± 0.06). In the healthy cows, *GST* had the highest potential amount of mRNA (1.87 ± 0.11), while *NCF4* had the lowest (0.39 ± 0.08).

## 4. Discussion

A better understanding of the genes, underlying mutations, and interactions with other factors that impart resistance is warranted in order to produce disease-resistant livestock or eradicate diseases [23]. The immune (*TLR4*, *TLR7*, *TNF-α*, *IL10*, *NCF4*, and *LITAF*), antioxidant (*ATOX1*, *GST*, and *OXSR1*), and erythritol-related (*TKT*, *RPIA*, and *AMPD1*) genes in endometritis-affected and healthy Holstein dairy cows were characterized in this research using a PCR-DNA sequencing technique. The findings show that the SNPs involving both categories vary. The chi-square study revealed that nucleotide polymorphism dispersion amongst the inspected calves was significant (*p* < 0.05). It is important to emphasize that the polymorphisms found and made available in this context provide additional data for the evaluated indicators when compared to the corresponding datasets acquired from GenBank.

There have been recent studies targeting novel genes specific to livestock endometritis susceptibility using genome-wide association analysis [32,41], but up to this point, no studies have examined the link between the SNPs in these genes and endometritis risk. The European cow (*Bos taurus*) gene sequences used in our study, which were reported in PubMed, are the first to demonstrate this association. According to our knowledge, there has not been any prior research on the variation of the immune (*TLR4*, *TLR7*, *TNF-α*, *IL10*, *NCF4*, and *LITAF*), antioxidant (*ATOX1*, *GST*, and *OXSR1*), and erythritol-related (*TKT*, *RPIA*, and *AMPD1*) markers and how they relate to postparturient endometritis in Holstein dairy cows. The candidate gene method, however, was employed to keep track of the soundness of endometritis-affected livestock. For example, endometritis and *CXCR1* SNPs have been linked in Holstein dairy cows [42]. In dairy cattle, uterine infection was linked to lactoferrin (*LTF*) gene polymorphism [43]. There has also been evidence linking the *beta defensin* gene polymorphism and clinical endometritis in dairy cows [44]. SNPs in the *TLR4* and *TLR2* genes and endometritis tolerance in buffalo have been elaborated [45,46]. 

The term “transcriptome” states that the genome’s complete set of genes that are reliably and efficiently expressed in various physiological and pathological conditions [19]. It is frequently used in inflammatory diseases to evaluate the immune system to identify pathogenic, diagnostic, and prognostic signatures, and has been helpful in the finding of novel therapeutic or diagnostic targets [47]. Through measuring the mRNA levels of immune (*TLR4*, *TLR7*, *TNF-α*, *IL10*, *NCF4*, and *LITAF*), antioxidant (*ATOX1*, *GST*, and *OXSR1*), and erythritol-related (*TKT*, *RPIA*, and *AMPD1*) genes, we examined the changes in the immune, redox, and erythritol metabolic state in postparturient endometritis-affected Holstein dairy cows compared with healthy ones. The *IL10*, *ATOX1*, and *GST* genes were expressed at significantly lower amounts in endometritis-affected cows according to the molecular changes. The expression levels of the genes *TLR4*, *TLR7*, *TNF-α*, *NCF4*, *LITAF*, *OXSR1*, *TKT*, *RPIA*, and *AMPD1* were significantly greater in endometritis-affected Holstein dairy cows than in resistant ones. Gene expression as well as genomic SNP markers were employed to evaluate genetic polymorphisms in order to address the limitations of earlier studies. The mechanisms that were investigated to regulate the immune, antioxidant, and erythritol-related indicators in both healthy and endometritis-affected cows are thus widely acknowledged. This is the first study to fully analyze the transcript levels of the immune, antioxidant, and erythritol-related indicators linked to the hazard of bovine endometritis. Consequently, qualitative and quantitative differences in the investigated genes’ expression precede the development of bovine uterine disease. 

Greater relative quantities of mRNA for the *IL1A*, *IL6*, *IL17A*, *TNF*, *PGES*, and *PGHS2* genes were found in primiparous Holstein cows postpartum when compared to healthy cows [48]. Additionally, *C3*, *C2*, *LTF*, *PF4*, and *TRAPPC13* had unique mRNA expression patterns in the blood and endometrial tissue of dairy cows with subclinical endometritis [49]. In contrast to control cows, cows with clinical and subclinical endometritis displayed a significant change in the mRNA expression of uterus-associated proinflammatory markers, according to Pothmann et al. [50]. In the endometrium of repeat breeding cows with and without subclinical endometritis, there were significantly more transcript levels of tumor necrosis factor and inducible nitric oxide synthase [51]. Three examined cytokines, including *IL-1*, *IL-1β*, and *IL-6*, were found to have increased gene expression in buffaloes with endometritis compared to healthy animals [52].

Innate immune systems, particularly Toll-like receptors and antimicrobial peptides, are vital for the endometrium’s first defense in contradiction of microorganisms [31,53]. Ten members of the Toll-like receptor (TLR) family are generally encoded in the mammalian genome, and they find pathogen-associated molecular forms [54]. TLR3, TLR7, TLR8, and TLR9 recognize nucleic acids and frequently from viruses, whereas TLR1, TLR2, and TLR6 distinguish bacterial lipids such as lipoteichoic acid (LTA) [31]. Lipopolysaccharide (LPS) from Gram-negative bacteria such as Escherichia coli is known by TLR4 [55]. TLR9 also categorizes bacterial DNA, and TLR5 binds bacterial flagellin [31]. The NOD1 and NOD2 receptors, which bind nucleotides, are used to detect bacteria that have entered host cells [31]. When TLRs are activated, proinflammatory mediators are formed, which then drive the immune response to rest the extent of the infections and eradicate them from the tissues [56]. To assist in removal of the pathogenic bacteria, for instance, TNF motivates the formation of antimicrobial peptides [57].

The anti-inflammatory cytokine IL-10 inhibits the production of natural killer cells IL-1 and TNF by macrophages, as well as IFN- and IL-2 by Th1 lymphocytes [58,59]. It has been demonstrated that IL-10 is formed by a diversity of T-cells and has anti-inflammatory possessions to defend uterine tissues from extremely virulent action of inflammatory cells and mediators through its interface with controlling suppressor T CD8+ cells, which could clarify its significant upsurge in uterine washings of cows with subclinical endometritis [60]. However, IL-10 has been shown to be the cause of inflammation, persistent post-partum infection, and damage to uterine local resistance [61]. The innate immunity gene neutrophil cytosolic factor 4 (NCF4) appears to be relevant and to be involved in the development of mastitis in cattle [62,63]. Lipopolysaccharide-induced TNF factor (LITAF) is a new protein that binds to a crucial area of the TNF promoter and is said to be responsible for activating *TNF-α* expression after LPS stimulation [64,65]. It has been proven that using the SNP in the *LITAF* gene in marker-assisted selection may increase chickens’ resistance to *Salmonella enteritidis* [66].

Oxidative stress markers have been linked to metabolic complications in recent years, especially in dairy cows where the peripartum period placed high loads on the body’s homeostatic routes [67]. According to research, the peripartum period’s antioxidant capability is insufficient to counteract the intensification in ROS [68]. As a result, the imbalance between increased ROS manufacture and decreased antioxidant fortifications close to parturition endorses oxidative stress and may be an issue in periparturient diseases in dairy cows [69]. Antioxidants protect by scavenging or detoxifying ROS, blocking their manufacturing, or sequestering transition metals that are the basis of free radicals [70]. Such mechanisms comprise both enzymatic and nonenzymatic antioxidant resistances formed within the body, known as endogenous antioxidant indicators as glutathione S transferase (GST) [71]. The *ATOX1* gene produces the copper metallochaperone protein recognized as ATOX1 [72]. ATOX1 guarded against reactive oxygen species in cells. As it transfers copper from the cytosol to transporters ATP7A and ATP7B, ATOX1 is essential for keeping copper homeostasis [73]. Serine/threonine protein kinase (OSR1) is encoded by the oxidative stress-responsive kinase 1 (*OXSR1*) gene, and it governs downstream kinases in reaction to environmental stressors [74]. When compared to the time at calving, the expression profile of *OXSR1* throughout the periparturient period exhibited a considerable up-regulation at (14) and (+14), with the lowest form seen at calving in dromedary camels [75].

Our study is the first to identify nucleotide sequence variations and the expression profile of genes related to erythritol in healthy Holstein dairy cows and those infected with endometritis. Our findings show that infected cows had higher mRNA values of *TKT*, *RPIA*, and *AMPD* than healthy ones. It has been established that erythritol has a reinforcing possible role in microbial virulence [76]; therefore, we utilized the genetic resistance to bovine endometritis found in the *TKT*, *RPIA*, and *AMPD* genes.

Multi-pathogen bacterial infections of the vaginal tract develop in dairy cattle following urination [7]. To remove pathogens from the uterus during bacterial infection, immune cells and endometrial cells provide a local immunological reaction [77]. A bacterial infection of the endometrium causes the production of chemokines and cytokines, which activates an inflammatory response. Leucocyte recruitment during inflammation has been reported to be interceded by inflammatory cytokines and complement fragments [78]. Endometritis is also characterized by unchecked extended inflammation linked to tissue damage, which causes the release of molecular forms accompanying injury, further aggravating inflammation and guaranteeing its perseverance [32]. Afterwards, oxidative stress is brought on by the extreme gathering of ROS [79]. These modifications are also associated with increased expression of molecules involved in LPS signaling, tissue remodeling, and acute phase response [49]. The aforementioned reasons could account for the significant amendment in the expression configuration of immune (*TLR4*, *TLR7*, *TNF-α*, *IL10*, *NCF4*, and *LITAF*), antioxidant (*ATOX1*, *GST*, and *OXSR1*), and erythritol-related (*TKT*, *RPIA*, and *AMPD1*) indicators in endometritis-affected cows. Thus, we assume that an infectious etiology is to blame for the bovine endometritis in the study’s cows. The endometritis-affected cows were exhibiting a substantial inflammatory response, as shown by our real-time PCR data. Gene expression disruption can be used to characterize the common pathological processes, whereas normal gene expression controls the bulk of physiological mechanisms [80,81]. Therefore, researching and classifying the genes that cause a phenotype should be possible through investigation of gene transcript level and the related molecular pathways.

## 5. Conclusions

Single nucleotide variants (SNPs) in the genes were discovered using PCR-DNA sequencing for immune (*TLR4*, *TLR7*, *TNF-α*, *IL10*, *NCF4*, and *LITAF*), antioxidant (*ATOX1*, *GST*, and *OXSR1*), and erythritol-related (*TKT*, *RPIA*, and *AMPD1*) genes found in resistant and endometritis-infected Holstein dairy cows. Additionally, healthy and affected cows showed differences in these markers’ mRNA amounts. By employing genetic markers accompanying natural resistance during cattle selection, these unique functional variants offer a promising chance to reduce bovine endometritis. Cows’ varying gene expression patterns for resistance and susceptibility to endometritis could function as guidance as well as an indicator for gauging their wellbeing. Forthcoming approaches to treating endometritis can be easily completed using the gene domains found here.

## Figures and Tables

**Figure 1 vetsci-10-00370-f001:**
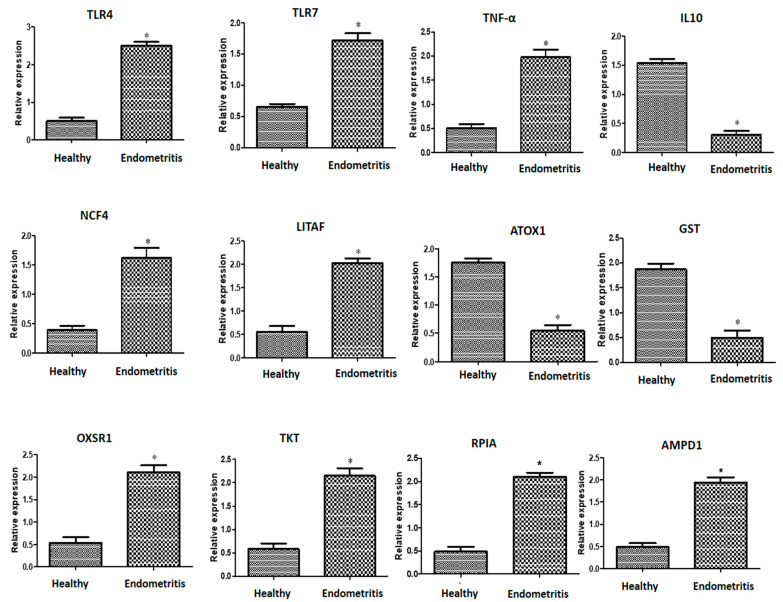
Differential transcript levels of immune, antioxidant, and erythritol-related genes between healthy (*n* = 65) and endometritis-affected (*n* = 65) dairy cows. The symbol * denotes significance when *p* < 0.05.

**Table 1 vetsci-10-00370-t001:** Oligonucleotide primer of immune, antioxidant, and erythritol-related genes employed for genetic polymorphisms.

Investigated Marker	Sense	Antisense	Annealing Temperature (°C)	Size of PCR Product (bp)	Reference
*TLR4*	5′-AGAGACGACACTACAGTGCCTCG-3′	5′-GAAGTCATTTAGAGAGACTAG-3′	60	528	Current study
*TLR7*	5′-TTTTCCACAGCTCATCTCTTCA-3′	5′-AAGGAGGCTGGAGAGATGCCTG-3′	60	420	Current study
*TNF-α*	5′-ACCAGCCAGGAGAGAGACAAGC-3′	5′-GTCAGCAGGCACCACCAGCTGGT-3′	62	551	Current study
*IL10*	5′-ATGCATAGCTCAGCACTACTCTG-3′	5′-TCACCATCCTGGAGGTCTTCT-3′	60	571	Current study
*NCF4*	5′-CCTGGGACCACAGCCTAAACGA-3′	5′-CTTGATGGTGCTGATGGTGTCC-3′	58	865	Current study
*LITAF*	5′-CTTTCTATGAAGGGCTTTTTTC-3′	5′-CACCCAATATACAGATTTTTGA-3′	62	644	Current study
*ATOX1*	5′-GCTCCTGTGGCGTGCACACCCG-3′	5′-TGGTGTGCAGGCCAAGACTTGG-3′	60	450	Current study
*GST*	5′-CGGCTCAGGCCGCCGCCGAGC-3′	5′-TGGGACAGCAGGGTCTCGAAAG-3′	58	480	Current study
*OXSR1*	5′-AGCGCCAGGCGCCGTCCGACC-3′	5′-GAGTATTGTAGCGATGGTAGC-3′	60	525	Current study
*TKT*	5′-GCTGCTTGCAGCTCCGCAGCC-3′	5′-GAGCCAGTGGCCACATCGGTGA-3′	62	456	Current study
*RPIA*	5′-CCACGTGCAGTTGCCGGGACGT-3′	5′-TTGATGAGGTTGAGGTCAGCGTC-3′	58	390	Current study
*AMPD1*	5′-CAGAGAGTGCAGATCACTGGC-3′	5′-ACTCATCCATCTCGTTGAGCAT-3′	60	502	Current study

*TLR4* = Toll-like receptor 4; *TLR7* = Toll-like receptor 7; *TNF-α =* tumor necrosis factor alpha; *IL10* = interleukin 10; *NCF4* = neutrophil cytosolic factor 4; *LITAF* = lipopolysaccharide-induced TNF factor; *ATOX1* = antioxidant 1 copper chaperone; *GST* = glutathione S-transferase; *OXSR1* = oxidative stress-responsive kinase 1; *TKT* = transketolase; *RPIA* = ribose 5-phosphate isomerase A; and *AMPD1* = adenosine monophosphate deaminase 1.

**Table 2 vetsci-10-00370-t002:** Oligonucleotide forward and reverse primers for immune, antioxidant and erythritol related genes under investigation used in real-time PCR.

Investigated Marker	Primer	Product Size (bp)	Annealing Temperature (°C)	GenBank Isolate	Origin
*TLR4*	F5′-CCTTGCGTACAGGTTGTTCC-3 R5′-GGCTGCCTAAATGTCTCAGGT-3′	133	59	MT424003.1	Current study
*TLR7*	F5′-CCAAGGTGCTTTCCAGTTGC-3 R5′-ACCAGACAAACCACACAGCA-3′	161	58	NM_001033761.1	Current study
*TNF-α*	F5′-AGAGACAAGCAGCTGCAGAAC-3′ R5′-GCAGGGTATGTGAGAGAGAGC-3′	96	60	NM_173966.3	Current study
*IL10*	F5′-GCACTACTCTGTTGCCTGGT-3′ R5′-AAGCTGTGCAGTTGGTCCTT-3′	179	60	NM_174088.1	Current study
*NCF4*	F5′-ATGAGGCGGGAGTTCCAGA-3′ R5′-CACCATGAGCTTCACGTCCT-3′	102	58	NM_001045983.1	Current study
*LITAF*	F5′-GCGGCGGTAAAATGTCTGTT-3′ R5′-TTGACAGCCACCGTCTCTTC-3′	100	58	NM_001046252.2	Current study
*ATOX1*	F5′-CAGGAAAGGCTGTCTCCTACC-3′ R5′-CCTAGATCTGTCTGGAGGGC-3′	116	59	NM_001130758.1	Current study
*GST*	F5′-ACCAGTCCAATGCCATCCTG-3′ R5′-CAGCGAAGGTCCTCTACACC-3′	115	60	NM_177516.1	Current study
*OXSR1*	F5′-CGCAGAGTAGCAAAGAGGCG-3′ R5′-CGCAAACTCACTGACCTCTCT-3′	187	59	NM_001075892.2	Current study
*TKT*	F5′-TGCTGAGATCATGGCTGTCC-3′ R5′-CCGTCCAAGTCGGAGTTGAT-3′	195	58	NM_001003906.1	Current study
*RPIA*	F5′-GAAGTCGACGCTGACCTCAA-3′ R5′-GGCAATCACGATGAAGCGAC-3′	99	59	NM_001035433.2	Current study
*AMPD1*	F5′-TTCGTCCAAAACCGCGTCTA-3′ R5′-TGAGGGTTGATGGTGGCTTC-3′	155	58	NM_001100349.1	Current study
*ß. actin*	F5′-TCGTGATGGACTCCGGTGA-3′ R5′-TGTCACGGACGATTTCCGCTC-3′	183	60	AY141970.1	Current study

*TLR4* = Toll-like receptor 4; *TLR7* = Toll-like receptor 7; *TNF-α =* tumor necrosis factor alpha; *IL10* = interleukin 10; *NCF4* = neutrophil cytosolic factor 4; *LITAF* = lipopolysaccharide-induced TNF factor; *ATOX1* = antioxidant 1 copper chaperone; *GST* = glutathione S-transferase; *OXSR1* = oxidative stress-responsive kinase 1; *TKT* = transketolase; *RPIA* = ribose 5-phosphate isomerase A; and *AMPD1* = adenosine monophosphate deaminase 1.

**Table 3 vetsci-10-00370-t003:** SNP distribution and kind of mutation for the genes under investigation in healthy and endometritis-affected Holstein cows.

Gene	SNPs	Healthy *n* = 65	Endometritis *n* = 65	Total *n* = 130	Kind of Inherited Change	Amino Acid Order and Sort	Chi Score	*p* Value
*TLR4*	T55C	38	-	38/130	Nonsynonymous	19 Y to H	60.95	<0.0001
	T171C	29	-	29/130	Synonymous	57 S	46.51	<0.0001
	G213A	-	41	41/130	Synonymous	71 Q	65.76	<0.0001
	G285T	48	-	48/130	Nonsynonymous	95 L to F	76.98	<0.0001
	C381T	35	-	36/130	Synonymous	127 D	57.74	<0.0001
	G400A	28	-	28/130	Nonsynonymous	134 D to N	44.91	<0.0001
	C491T	-	37	37/130	Nonsynonymous	164 A to V	59.34	<0.0001
*TLR7*	G56A	42	-	42/130	Nonsynonymous	19 C to Y	67.37	<0.0001
*TNF-α*	T87C	26	-	26/130	Synonymous	29 L	41.70	<0.0001
	T208C	-	31	31/130	Nonsynonymous	70 C to R	49.72	<0.0001
	A389G	-	52	52/130	Nonsynonymous	130 K to R	83.40	<0.0001
*IL10*	G148A	-	30	30/130	Nonsynonymous	50 E to K	48.12	<0.0001
	C152T	55	-	55/130	Nonsynonymous	51 A to V	88.22	<0.0001
	G225A	34	-	34/130	Synonymous	75 K	54.53	<0.0001
	G321A	-	28	28/130	Synonymous	107 E	44.89	<0.0001
	G357C	49	-	49/130	Synonymous	119 L	78.59	<0.0001
*NCF4*	A744G	-	37	37/130	Synonymous	248 P	59.34	<0.0001
*LITAF*	A392G	56	-	56/130	Nonsynonymous	131 D to G	89.82	<0.0001
*ATOX1*	A75G	-	27	27/130	Synonymous	25 A	43.31	<0.0001
	C141T	-	46	46/130	Synonymous	47 C	73.78	<0.0001
*GST*	A30G	36	-	36/130	Synonymous	10 E	57.74	<0.0001
*OXSR1*	G35T	48	-	48/130	Nonsynonymous	12 R to L	76.98	<0.0001
	C195T	21	-	21/130	Synonymous	65Y	33.68	<0.0001
	A270G	30	-	30/130	Synonymous	90 K	48.12	<0.0001
	C414A	-	51	51/130	Synonymous	139 V	81.80	<0.0001
*TKT*	G76T	-	39	39/130	Nonsynonymous	26 G to W	62.55	<0.0001
	A396G	53	-	53/130	Synonymous	132 Q	85.01	<0.0001
*RPIA*	G56A	33	-	33/130	Nonsynonymous	19 R to H	52.93	<0.0001
	T72C	44	-	44/130	Synonymous	24 H	70.57	<0.0001
	C202T	-	57	57/130	Nonsynonymous	68 R to C	91.43	<0.0001
*AMPD1*	T315C	47	-	47/130	Synonymous	105 N	75.39	<0.0001

*TLR4* = Toll-like receptor 4; *TLR7* = Toll-like receptor 7; *TNF-α =* tumor necrosis factor alpha; *IL10* = interleukin 10; *NCF4* = neutrophil cytosolic factor 4; *LITAF* = lipopolysaccharide-induced TNF factor; *ATOX1* = antioxidant 1 copper chaperone; *GST* = glutathione S-transferase; *OXSR1* = oxidative stress-responsive kinase 1; *TKT* = transketolase; *RPIA* = ribose 5-phosphate isomerase A; and *AMPD1* = adenosine monophosphate deaminase 1. A = alanine; C = cysteine; D = aspartic acid; E = glutamic acid; F = phenylalanine; G = glycine; H = histidine; K = lysine; L = leucine; N = asparagine; P = proline; Q = glutamine; R = arginine; S = serine; V = valine; W = tryptophan; and Y = tyrosine.

## Data Availability

Upon reasonable request, the supporting information for the study’s findings will be provided by the corresponding author.

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
