# Peer review of "Investigating Polymorphisms and Expression Profile of Immune, Antioxidant, and Erythritol-Related Genes for Limiting Postparturient Endometritis in Holstein Cattle"

_vetsci, 2023, doi:10.3390/vetsci10060370_

Round 1

Reviewer 1 Report

The data was well-presented. The paper was well-written. The paper could be accepted after the following minor revision. 

Accepted after minor editing of English language

Author Response

Reviewer 1

Comments and Suggestions for Authors

Comment

The data was well-presented. The paper was well-written. The paper could be accepted after the following minor revision. 

Response

We thank reviewer for this positive comment.

Comment

Comments on the Quality of English Language

Accepted after minor editing of English language

Response

We are grateful to the reviewer for drawing it to our consideration. Minor English editing is made.

Reviewer 2 Report

This study presents a comprehensive investigation of genetic polymorphisms and transcript levels of immune, antioxidant and erythritol related markers for predicting and tracking postparturient endometritis in Holstein dairy cows. The outcomes of this study may contribute to developing a workable control plan for postparturient endometritis in dairy cows. Therefore, the reviewer would like to support the publication of this paper.

Specific comments:

1. The association between erythritol-related genes and cow endometritis should at the very least be explained in the article's introduction.

2. The author should clearly state the number of samples/repeats used, as well as the value of n, in the legend of Figure 1 in order to make the description of the approach more understandable.

3.  Clearly state the time frame for sample collection in Method and Materials 2.1, e.g., xx days postpartum.

4. Supplementary image S5 needs to have its image resolution increased because it is somewhat fuzzy.

Author Response

Reviewer 2

Comment

This study presents a comprehensive investigation of genetic polymorphisms and transcript levels of immune, antioxidant and erythritol related markers for predicting and tracking postparturient endometritis in Holstein dairy cows. The outcomes of this study may contribute to developing a workable control plan for postparturient endometritis in dairy cows. Therefore, the reviewer would like to support the publication of this paper.

Response

We thank reviewer for this positive comment.

Specific comments:

Comment

  1. The association between erythritol-related genes and cow endometritis should at the very least be explained in the article's introduction.

Response

We are grateful to the reviewer for drawing it to our consideration. The association between erythritol-related genes and cow endometritis is explained in the article's introduction. Actually our study is he first that associate erythritol-related genes and cow endometritis; However we add additional information.

Comment

  1. The author should clearly state the number of samples/repeats used, as well as the value of n, in the legend of Figure 1 in order to make the description of the approach more understandable.

Response

We thank reviewer for this. The number of samples/repeats used, as well as the value of n, in the legend of Figure 1 is made in order to make the description of the approach more understandable.

Comment

  1. Clearly state the time frame for sample collection in Method and Materials 2.1, e.g., xx days postpartum.

Response

We are grateful to the reviewer for drawing it to our consideration. The time frame for sample collection is mentioned.

Comment

  1. Supplementary image S5 needs to have its image resolution increased because it is somewhat fuzzy.

Response

We are grateful to the reviewer for drawing it to our consideration. Supplementary image S5 resolution is increased.

Reviewer 3 Report

Introduction

Use this definition of transcriptome: A transcriptome is the full range of messenger RNA, or mRNA, molecules expressed by an organism.

2.1

Edit: This research used  Holstein dairy cows (n=130) reared on a private farm in the area of Ismailia, Egypt.... 

Edit: The endometritis cows were selected based on body temperature and  physical examination findings postpartum.

Edit: The animals were examined and findings (body's temperature, pulse, respiration rate, mucous membranes, and vag-inal discharge) recorded  [29].   

Discussion

Edit: An better understanding of the genes,  underlying mutations, and interactions with other factors that impart resistance is warranted in order to produce disease-resistant livestock or  eradicate diseases [

Minor revisions required.

Author Response

Reviewer 3

Comment

Introduction

Use this definition of transcriptome: A transcriptome is the full range of messenger RNA, or mRNA, molecules expressed by an organism.

Response

We thank the reviewer for this. This definition of transcriptome is used.

Comment

2.1

Edit: This research used Holstein dairy cows (n=130) reared on a private farm in the area of Ismailia, Egypt.... 

Response

We thank the reviewer for this. The sentence is edited.

Comment

Edit: The endometritis cows were selected based on body temperature and physical examination findings postpartum.

Response

We are grateful to the reviewer for drawing it to our consideration. The sentence is edited.

Comment

Edit: The animals were examined and findings (body's temperature, pulse, respiration rate, mucous membranes, and vag-inal discharge) recorded [29].   

Response

We thank the reviewer for this. The sentence is edited.

Discussion

Comment

Edit: A better understanding of the genes, underlying mutations, and interactions with other factors that impart resistance is warranted in order to produce disease-resistant livestock or eradicate diseases [

Response

Many cardinal thanks for the reviewer. The sentence is edited.

Comment

Comments on the Quality of English Language

Minor revisions required.

Response

We are grateful to the reviewer for drawing it to our consideration. Minor English editing is made. All suggestions have been made.